# Development of a core outcome set for congenital pulmonary airway malformations: study protocol of an international Delphi survey

Sergei Hermelijn  ,[1] Casper Kersten,[1] Dhanya Mullassery,[2] Nagarajan Muthialu,[2] Nazan Cobanoglu,[3] Silvia Gartner,[4] Pietro Bagolan,[5] Carmen Mesas Burgos,[6] Alberto Sgro,[7] Stijn Heyman,[8] Holger Till,[9] Janne Suominen,[10] Maarten Schurink,[11] Liesbeth Desender,[12] Paul Losty,[13] Kjetil Ertresvag,[14] Harm A W M Tiddens,[15] Rene M H Wijnen,[1] Marco Schnater,[1] On behalf of the CONNECT study consortium COS development group, On behalf of the CONNECT study consortium COS development group

For numbered affiliations see end of article.

**Correspondence to**
Dr Marco Schnater;
j.schnater@erasmusmc.nl

## ABSTRACT

**Introduction** A worldwide lack of consensus exists on the optimal management of asymptomatic congenital pulmonary airway malformation (CPAM) even though the incidence is increasing. Either a surgical resection is performed or a wait-and-see policy is employed, depending on the treating physician. Management is largely based on expert opinion and scientific evidence is scarce. Wide variations in outcome measures are seen between studies making comparison difficult thus highlighting the lack of universal consensus in outcome measures as well. We aim to define a core outcome set which will include the most important core outcome parameters for paediatric patients with an asymptomatic CPAM.

**Methods and analysis** This study will include a critical appraisal of the current literature followed by a three-stage Delphi process with two stakeholder groups. One surgical group including paediatric as well as thoracic surgeons, and a non-surgeon group including paediatric pulmonologists, intensive care and neonatal specialists. All participants will score outcome parameters according to their level of importance and the most important parameters will be determined by consensus.

**Ethics and dissemination** Electronic informed consent will be obtained from all participants. Ethical approval is not required. After the core outcome set has been defined, we intend to design an international randomised controlled trial: the COllaborative Neonatal NEtwork for the first CPAM Trial, which will be aimed at determining the optimal management of patients with asymptomatic CPAM.

## Strength and limitations of this study

► The core outcome set is a disease-specific collection of the most important outcomes that will be established by consensus between key stakeholders. Participants will be an international group of specialists with experience in the treatment of patients with congenital pulmonary airway malformation (CPAM), which will result in a universally approved synthesis of expert opinion.

► This protocol describes an international online Delphi survey that should identify the most important core outcome parameters including optimal timing and modality to monitor paediatric patients with an asymptomatic CPAM.

► Existing comprehensive literature reviews shall be used to inform initial outcome parameters instead of a systematic literature search as no randomised trials and very few prospective studies have yet been published regarding the outcome of patients with CPAM.

► Parents' and patients' views are not included in the protocol as the final core outcome set is specifically intended for patients with asymptomatic CPAM, however their input will play an important role when designing future studies.

## INTRODUCTION

Congenital pulmonary airway malformation (CPAM), formerly known as congenital cystic adenomatoid malformation, is the most common congenital lung abnormality.[1] The incidence of CPAM has increased up to 4 per 10 000 births over the last years.[2] Most infants born with a CPAM are asymptomatic but others may show symptoms such as neonatal respiratory distress, persistent cough or recurrent lung infections in the first years of life.[3] A worldwide lack of consensus exists on the optimal management and follow-up of infants with an asymptomatic CPAM.[4 5] Prospective studies on postnatal management are lacking, and cohort studies vary widely in the outcome measures they report.[2 6]

An asymptomatic CPAM is either surgically resected or a wait-and-see policy is employed depending on physician preference or local guidelines. Either way, cases are ideally discussed in a multidisciplinary team in which parental preferences are taken into account as well. To date, no definitive evidence exists on the optimal management, long-term outcomes are still unknown and factors for predicting symptoms are still being investigated.[6] The arguments for a wait-and-see policy are that the malformation is originally benign, potentially regresses and in most cases remains asymptomatic.[7] The arguments for surgical management include the risk of recurrent lung infections (which could make subsequent surgery more difficult), risk of acute respiratory distress, potential malignant transformation, parental anxiety and allowing for compensatory lung growth.[8 9]

Consensus needs to be reached on outcome measures and their timing that can be applied in international studies aimed at identifying the optimal management of asymptomatic CPAM. A core outcome set (COS) is a disease-specific collection of outcomes that have been identified by consensus between key stakeholders as being the most important in determining success of a treatment.[10] Such consensus is often reached through a Delphi method in which stakeholders anonymously rate outcome measures according to their importance, in one or multiple rounds.[11] We aim to develop a COS for patients with CPAM using the Delphi method as a tool for reaching consensus and present the study protocol for this.

## METHODS

The COS development will follow the Core Outcome Set-STAndards for Development Recommendations[12] and the Core Outcome Measures in Effectiveness Trials (COMET) handbook.[11] This COS development study was registered with the COMET Initiative in May 2020 (http://www.comet-initiative.org/studies/details/1570).

### Scope

This protocol describes the Delphi method which shall define a COS for all asymptomatic patients who are either prenatally or postnatally diagnosed with a CPAM. Asymptomatic patients are defined as those who have no need for prolonged respiratory support (>24 hours) including supplemental oxygen and ventilation. The COS will include the most important outcome measures for patients with CPAM, regardless of the management. The COS may be used as a guideline for clinical follow-up or in future research studies. After the COS has been defined, we intend to design a randomised controlled trial: the COllaborative Neonatal NEtwork for the first European CPAM Trial (CONNECT). This trial will be performed by the CONNECT study consortium and aims to identify the optimal management of patients with asymptomatic CPAM. The COS is intended for paediatric patients up until the age of 18 years and will not include fetal outcome. Outcome parameters as well as their measurement instruments, and age at assessment will be included.

### Study design

The COS will be developed in an online, three-round, Delphi process, preceded by an appraisal of previously published literature.

### Literature review

A systematic literature review is recommended to yield an initial outcome set for the first round in the Delphi process.[11] To date, no randomised studies have yet been done and other studies examining the management of asymptomatic CPAMs report a large variety in outcome parameters.[6] A literature review will be done by the study management coordinators (SH, CK) informed by two literature reviews, previously published in a special issue of a paediatric surgery scientific journal, each covering opposing thoughts on the arguments for surgical management[8] or a wait-and-see policy.[7] In addition, we will scrutinise a recent systematic review and meta-analysis, which covers the risks associated with either surgical resection or a wait-and-see policy of asymptomatic CPAM.[3] Existing definitions, measurement tools and common measurement time-points for outcomes will be extracted, and formatted into appropriately phrased questions for use in the first round of the Delphi process. The independent coordinators will be blinded for participant identity during the process by means of a unique identification number and will ensure the Delphi process is performed according to the protocol.

### Stakeholders and recruitment

Paediatric surgeons and thoracic surgeons are the healthcare professionals who are most frequently involved in the operative management of patients with CPAM. Initial consultation or follow-up is variably done by paediatric surgeons, maternal-fetal medicine specialists, neonatologists, paediatricians or paediatric pulmonologists. We therefore decided to form two stakeholder groups: (1) surgeons (ie, paediatric and thoracic surgeons) and (2) non-surgeons (eg, maternal-fetal medicine specialists, paediatricians, paediatric pulmonologists and neonatologists).

We will recruit study participants through an existing international network of paediatric surgeons and pulmonologists who have expressed interest in collaborating on the CONNECT trial. We will inform potential participants on the aims and procedures of the Delphi process and encourage them to enrol other specialists involved in the care of patients with CPAM in their own centres. Prior to the first round, those who have been found willing to participate will be sent an email explaining the aims and procedures of the Delphi process, and emphasising the importance of finishing each round within the allocated time.

 Hermelijn S, *et al. BMJ Open* 2021;**11**:e044544. doi:10.1136/bmjopen-2020-044544

## Sample size

There is no consensus on the optimal sample size for a Delphi study[11]; recruitment will therefore be based on the prospective study for which it is primarily intended (CONNECT). For the CONNECT trial, we aim to include at least 12 international centres, and therefore set the minimum of participants in each stakeholder group in the final round at 12. To reduce bias, no more than two participants from a single centre can participate in a stakeholder group. To minimise attrition bias in consecutive rounds, we aim to achieve that 75% of participants complete a round.[11 13] Therefore, the minimum number of participants in each stakeholder group for round 1 will be 21, and 16 for round 2. We believe that these minimum numbers constitute a representative sample, considering the rarity of the disease and the limited number of professionals with experience in managing this disease. Previous Delphi COS studies in paediatric surgery used similar numbers for investigating a more common disease such as appendicitis.[14 15]

## Attrition bias

Attrition bias will be assessed separately for each round of the Delphi process, and separately in each stakeholder group. In each group, the median score of every outcome will be compared using the Wilcoxon rank-sum test, between those only completing the previous round and those completing the consecutive round as well.[11]

## Delphi study
### Consensus

Participants will be asked to score each outcome parameter using the Grading of Recommendations, Assessment, Development and Evaluations scale.[16] The 9-point Likert scale will label 1–3 as 'not important', 4–6 as 'important but not critical' and 7–9 as 'critical'. Consensus for inclusion is reached if ≥70% of participants rate the outcome parameter 7–9 and <15% rate it 1–3. Consensus for exclusion is reached if >70% participants rate the outcome 1–3 and <15% rate it 7–9. Outcomes not meeting these definitions will be classified as 'no consensus'.[11]

### Timeline

Participants will be asked to complete each round of the Delphi process within 4 weeks. A weekly reminder email will be sent to those who have not yet completed the survey. Those failing to complete the questionnaire within the allocated 4 weeks will be excluded from next rounds. The deadline shall be extended if the projected minimum sample size has not been reached and those failing to complete the questionnaire shall be approached individually.

### Delphi round 1

The three-round Delphi process shall be performed using 'Welphi', an online data system specifically developed for this use.[17] All participants shall be approached simultaneously and asked to rate each of the previously identified outcome parameters on importance as follows:

*"How important would you rate the following outcome parameter including measurement instrument and age in determining the best management of asymptomatic CPAM patients?"*

Participants are invited to suggest additional outcome parameters stating: (1) the outcome parameter, (2) the measurement instrument and (3) the age at assessment. These additional outcome parameters will be scored in the second round.

### Delphi round 1 analysis

Outcome parameters will be analysed separately for each stakeholder group (surgeons and non-surgeons) and all parameters will be included in the second round of the Delphi process. The additional outcomes provided by participants will be reviewed to confirm they represent new outcomes. If confirmed, the item in question will be included in the second round as well.

### Delphi round 2

All participants who completed the first round will automatically be invited to participate in round 2.

Per stakeholder group, the median scores assigned in the first round will be made known. This will allow participants to consider the views of the other participants in the stakeholder group. They will be invited to look at all items again and consider adjusting their own scores. Furthermore, they will be asked to score the newly added outcome parameters suggested in the first round.

### Delphi round 2 analysis

All outcome parameters meeting consensus criteria for exclusion by all participants will be excluded from the third round. All other parameters will be included in the third round.

### Delphi round 3

All participants who completed the first and second rounds will be invited to participate in the third round. The median score of their own stakeholder group and the score of the other stakeholder group will be presented to participants. This will allow participants to consider the views of the other stakeholder group before rescoring the outcomes. They will be invited to look at all remaining items again and consider adjusting their own scores. In addition, participants will be asked to identify a single outcome parameter, which is the most important for determining the treatment choice in patients with asymptomatic CPAM according to them.

### Delphi round 3 analysis and final COS development

All outcome parameters meeting the criteria for consensus of inclusion by all participants will be included in the final COS. All other outcome parameters will be excluded. To achieve a COS that is feasible for clinical use in trials, we aim to include a maximum of 10 outcome parameters in the final COS. If the number of outcome parameters meeting the criteria for consensus of inclusion greatly exceeds this maximum number, we will only include the 10 outcomes with the highest level of consensus for the

COS and report those excluded in this stage. The level of consensus will be determined by the median score of each outcome parameter in round 3.

The final COS will be a collection of the most important outcome parameters in patients with CPAM. The final COS will be annotated according to the outcome taxonomy, which was constructed to maximise future data harmonisation. Additionally, the final COS will be divided into the four core areas of the OMERACT filter: death, life impact, pathophysiological manifestations and resource use.[10]

### Ethics and dissemination
Electronic informed consent will be obtained from all participants. Prior ethical approval for the Delphi study is not required. The final COS will be published in an international peer-reviewed scientific journal and on the COMET Initiative website (https://www.comet-initiative. org/).

### Data collection and confidentiality
Participants will complete questionnaires using the 'Welphi' survey tool.[17] Anonymised data will be stored on a secure online server and will be managed according to the European General Data Protection Regulation.[18]

**Author affiliations**
[1]Pediatric Surgery, Erasmus MC Sophia Children's Hospital, Rotterdam, The Netherlands
[2]Pediatric Surgery, Great Ormond Street Hospital for Children, London, UK
[3]Pediatric Pulmonology, Ankara University Faculty of Medicine, Ankara, Turkey
[4]Pediatric Pulmonology, Hospital Universitari Vall d'Hebron, Barcelona, Spain
[5]Department of Medical and Surgical Neonatology, Ospedale Pediatrico Bambino Gesu, Roma, Italy
[6]Pediatric Surgery, Karolinska Institutet, Stockholm, Sweden
[7]Pediatric Surgery, Padua University Hospital, Padova, Italy
[8]Pediatric Surgery, ZNA-GZA Paola Children's Hospital, Antwerp, Belgium
[9]Pediatric Surgery, Medical University of Graz, Graz, Austria
[10]Pediatric Surgery, University of Helsinki Children's Hospital, Helsinki, Finland
[11]Pediatric Surgery, Radboud University Medical Centre Amalia Children's Hospital, Nijmegen, The Netherlands
[12]Pediatric Surgery, Ghent University Faculty of Medicine and Health Sciences, Gent, Belgium
[13]Pediatric Surgery, University of Liverpool, Liverpool, UK
[14]Pediatric Surgery, Oslo University Hospital, Oslo, Norway
[15]Pediatric Pulmonology, Erasmus MC Sophia Children's Hospital, Rotterdam, The Netherlands

**Collaborators** The following people form the CONNECT study consortium COS development group: S.M. Hermelijn, C.M. Kersten, J.M. Schnater, R.M.H. Wijnen, H.A.W.M. Tiddens, S.C.M. Cochius-den Otter (Sophia Children's Hospital, Rotterdam, The Netherlands); J. Suominen, M. Pakarinen, L. Martelius (Helsinki Children's Hospital, Helsinki, Finland); S. Heyman, D. Vervloessem (ZNA-GZA Paola Children's Hospital, Antwerp, Belgium); H. Steyaert (Queen Fabiola Children's University Hospital, Brussels, Belgium); A. Sgro, P. Gamba (University Hospital Padua, Padua, Italy); M. Schurink, S. van der Heide, J. Roukema (Amalia Children's Hospital, Nijmegen, The Netherlands); N. Rikkers-Mutsaerts (Leiden University Medical Centre, Leiden, The Netherlands); S. Terheggen-Lagro, S. de Beer, E. Haarman (Amsterdam University Medical Centre, Amsterdam, The Netherlands); H. Till, G. Singer (Medical University of Graz, Graz, Austria); M. Metzelder, P. Sezen (Medizinische Universität Wien, Vienna, Austria); L. Desender, H. Schaballie (Ghent University Hospital, Ghent, Belgium); N. Cobanoglu, G. Gollu (Ankara University, School of Medicine, Ankara, Turkey); M. Stanton (University Hospital Southampton, Southampton, UK); A. Bonnard (Hopital Universitaire Robert Debré, Paris, France); R. Sfeir (Hopital Jeanne de Flandre, Lille, France); N. Muthialu, D. Mullassery, C. Wallis (Great Ormond Street Hospital, London, UK); D. Cox (Children's Health Ireland at Crumlin, Dublin, Ireland); P. Bagolan, F. Morini (Bambino Gesu Pediatric Hospital, Rome, Italy); C. Mesas Burgos, P. Conner, E. Caffrey Osvald, C. Bitkover (Karolinska Institutet, Stockholm, Sweden); H. Decaluwe, M. Proesmans, M. Boon, J. Deprest (University Hospital Leuven, Leuven, Belgium); S. Gartner, A. Lain (Hospital Universitari Vall d Hebron, Barcelona, Spain); P.D. Losty, I. Sinha (Alder Hey Children's Hospital, School Of Health and Life Science, University of Liverpool, Liverpool, UK); I. Yardley (Evelina London Children's Hospital, London, UK); M. Singh (Birmingham Children's Hospital, Birmingham, UK); L. Wessel, K. Zahn, T. Schaible (University Hospital Mannheim, Mannheim, Germany); N. Qvist (Odense Univeristy Hospital, Odense, Denmark); M. Zampoli (Red Cross Children's Hospital, Cape Town, South Africa); G. Aksnes (Oslo University Hospital, Oslo, Norway); C.K. van der Ent, K.M. Winter-de Groot (University Medical Centre Utrecht, Utrecht, The Netherlands); R. Peters (Royal Manchester Children's Hospital, Manchester, UK); E. Hannon (Leeds Children's Hospital, Leeds, UK); Q. Jöbsis, M. Bannier (Maastricht University Medical Centre, Maastricht, The Netherlands).

**Contributors** All authors contributed to the design of this protocol. SH, DM, NC, SG, PB, CMB, AS, SH, JS, MS, LD, KE, HT, RW and JMS initiated the project. The protocol was drafted by SH and CK. The protocol was critically reviewed by DM, NM, NC, SG, PB, CMB, AS, SH, HT, JS, MS, LD, PL, KE, HAWMT, RMHW and JMS. All authors contributed to the manuscript and read and approved the final manuscript. The CONNECT study consortium COS development group consists of all participants of the Delphi process. They have all read, refined and approved the final manuscript.

**Funding** The authors have not declared a specific grant for this research from any funding agency in the public, commercial or not-for-profit sectors.

**Competing interests** None declared.

**Patient and public involvement** Patients and/or the public were not involved in the design, or conduct, or reporting, or dissemination plans of this research.

**Patient consent for publication** Not required.

**Provenance and peer review** Not commissioned; externally peer reviewed.

**ORCID iD**
Sergei Hermelijn http://orcid.org/0000-0002-7296-9932

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
