## [Reviewer comments · BMJ Open]

ARTICLE DETAILS

TITLE (PROVISIONAL)	Development of a Core Outcome Set for Congenital Pulmonary Airway Malformations: study protocol of an international Delphi survey
AUTHORS	Hermelijjn, Sergei; Kersten, Casper; Mullassery, Dhanya; Muthialu, Nagarajan; Cobanoglu, Nazan; Gartner, Silvia; Bagolan, Pietro; Mesas Burgos, Carmen; Sgro, Alberto; Heyman, Stijn; Till, Holger; Suominen, Janne; Schurink, Maarten; Desender, Liesbeth; Losty, Paul; Ertresvag, Kjetil; Tiddens, Harm A. W. M.; Wijnen, Rene; Schnater, Marco

VERSION 1 – REVIEW

REVIEWER	Shaun Kunisaki Johns Hopkins University, USA
REVIEW RETURNED	14-Oct-2020

GENERAL COMMENTS	This is a paper that proposes a study protocol for the establishment of an international randomized trial of operative vs. non-operative management of CPAM. The delphi method is proposed. The proposed attempt at an RCT to study is important question should be applauded. The manuscript is clearly written. However, its impact appears to be very incremental for the typical BMJ open reader, given the absence of any results and the rare nature of these lesions. My concerns are: 1) The authors do not fairly discuss the controversy and lack of equipoise of this trial in North America (esp. the USA). Such a study would not be feasible in the US given the clinical outcomes data on surgical management, esp. in the era of thoracoscopic surgery. The controversy needs to be acknowledged by this largely European group of investigators.2) The inclusion criteria needs to be better defined. Are these authors talking about all CPAMs or just ones that are prenatally diagnosed. I think there is more consensus on the latter. Postnatally diagnosed macrocystic CPAMs can sometimes be confused for pleuropulmonary blastoma. This important fact needs to be acknowledged.3) It would be helpful if the outcome measures of interest could be more clearly stated in the manuscript. How these lesions should be followed long-term is still unclear and the study endpoint is also problematic.
---

REVIEWER	Berman, Loren Nemours Children's Hospital
REVIEW RETURNED	16-Oct-2020

GENERAL COMMENTS	This manuscript describes a protocol for a Delphi method study to develop a core outcome set to be used in an RCT evaluating the approach to management of patients with asymptomatic CPAM. I am not accustomed to reviewing study protocols for publication, but my overall comment is that i would suggest completing the systematic literature review and including the content for Delphi method process in the protocol publication. In the current format, there is not much substance to this protocol so it is difficult to critique. After reading the protocol, it is unclear what the proposed core outcome set might look like. For example, is the primary goal to come up with an approach to imaging that would be used if patients were randomized to have observation versus surgery? Or are asymptomatic patients being randomized to different observation protocols? Is part of the goal of the Delphi method study to generate inclusion/exclusion criteria for the RCT? Define what is meant by symptomatic vs asymptomatic? none of this is clearly defined. It would be helpful if the authors could elaborate on the plans for the RCT as this would help the reader to conceptualize what the COS might look like. Right now it is unclear. On a related note, if the COS is going to include imaging surveillance plan for patients randomized to non-operative management, it seems like radiologists should be included in stakeholder group and I did not see them mentioned. I am very excited to see that development of an RCT to answer questions about the optimal approach to management of children with asymptomatic CPAM is underway!
--

REVIEWER	Hatem Beshir 1. Department of Cardiothoracic Surgery, Faculty of Medicine, Mansoura University, Egypt 2. Department of Cardiothoracic Surgery, Ministry of Health of Egypt, Alexandria Directorate, Alexandria, Egypt.
REVIEW RETURNED	21-Oct-2020

GENERAL COMMENTS	I would like to thank the editor for giving me the privilege to review the manuscript entitled: "Development of a Core Outcome Set for Congenital Pulmonary Airway Malformations: study protocol of an international Delphi survey". Congratulations for a well written protocol and great effort on a very important and a debatable segment in thoracic surgery spectrum - Management of congenital pulmonary airway malformations. The working group have chosen the Delphi technique where a group of experts exchange views, and each independently gives estimates and assumptions to a facilitator who reviews the data and issues a summary report. The working group has divided the expert teams into two groups to review the diversity in the management within surgeons and non-surgeons. The results of this study would be included in a clinical trial. It is written using good English, I believe it is satisfactory and suitable for publication pending minor revisions. Kindly note the following corrections I would request you to make to improve the readability of your manuscript.
---

	1. The abstract: I recommend to a conduct systematic review of all relevant literature according to the Preferred Reporting Items for Systematic Reviews and Meta-Analyses (PRISMA) Statement. Retrospective studies and non-randomized trials are considered evidence that should be mentioned. 2. Please report your literature search strategy. Kindly mention Study selection criteria. Please report that you will only include randomized controlled trials and explain why would you exclude non RCT's and cohort studies. Thank you. 3. All relevant outcome parameters identified from the above-mentioned reviews will be used in the first round of “the” Delphi process. Please identify and mention the role of the facilitator. In case of disagreement, please mention how would this be resolved. Thank you. 4. Please mention that the outcomes will be formatted into questions and taken forward to the Delphi consensus exercise. To ensure appropriate phrasing and understanding for all stakeholders. Thank you. 5. Would the participants who registered for the survey be given a unique number to allow the Study Management Group to track attrition? 6. Please report the duration of each Delphi round. What is the policy of management of non-respondents? When and how would they be notified electronically? 7. Please report data extraction method and analysis. 8. Please report the current trail status. Thank you. 9. Please revise the referencing style for references 17 and 18. 10. Please check 2 typos. (first round ofthe -- (1) Surgeons (e.i. pediatric --) I would like to thank the authors for their effort and I recommend the manuscript to be accepted after minor revisions required.
--	--

VERSION 1 – AUTHOR RESPONSE

Reviewer: 1

This is a paper that proposes a study protocol for the establishment of an international randomized trial of operative vs. non-operative management of CPAM. The delphi method is proposed. The proposed attempt at an RCT to study is important question should be applauded. The manuscript is clearly written. However, its impact appears to be very incremental for the typical BMJ open reader, given the absence of any results and the rare nature of these lesions.

My concerns are:

1) The authors do not fairly discuss the controversy and lack of equipoise of this trial in North America (esp. the USA). Such a study would not be feasible in the US given the clinical outcomes data on surgical management, esp. in the era of thoracoscopic surgery. The controversy needs to be acknowledged by this largely European group of investigators.

We agree with the reviewer that a more in depth discussion of the controversy surrounding the management of asymptomatic CPAM is relevant. However, we believe this discussion may be more relevant in the final outcome set as this is the protocol for the Delphi study we will be performing. The outcome of this Delphi study will be a core outcome set which can be used for clinical follow-up or in future research. The trial will be designed after this process in which the final core outcome set as well as specific issues regarding a trial design will be addressed. In the “Scope” section we have better specified the purpose of this protocol for the Delphi study.

2) The inclusion criteria needs to be better defined. Are these authors talking about all CPAMs or just ones that are prenatally diagnosed. I think there is more consensus on the latter. Postnatally diagnosed macrocystic CPAMs can sometimes be confused for pleuropulmonary blastoma. This important fact needs to be acknowledged. We agree with the reviewer that the distinction with PPB is indeed crucial, especially in postnatally diagnosed cases. We have stated in the “Scope” section that this Core Outcome Set will include postnatal outcome parameters in patients who are either prenatally or postnatally diagnosed.

3) It would be helpful if the outcome measures of interest could be more clearly stated in the manuscript. How these lesions should be followed long-term is still unclear and the study endpoint is also problematic.

We agree with the reviewer that this information is indeed helpful to interpret the final core outcome set. We intend to elaborate on this matter in the outcome of the Delphi study.

Reviewer: 2

This manuscript describes a protocol for a Delphi method study to develop a core outcome set to be used in an RCT evaluating the approach to management of patients with asymptomatic CPAM. I am not accustomed to reviewing study protocols for publication, but my overall comment is that I would suggest completing the systematic literature review and including the content for Delphi method process in the protocol publication. In the current format, there is not much substance to this protocol so it is difficult to critique.

After reading the protocol, it is unclear what the proposed core outcome set might look like. For example, is the primary goal to come up with an approach to imaging that would be used if patients were randomized to have observation versus surgery? Or are asymptomatic patients being randomized to different observation protocols? Is part of the goal of the Delphi method study to generate inclusion/exclusion criteria for the RCT? Define what is meant by symptomatic vs asymptomatic? None of this is clearly defined.

We agree with the reviewer that the purpose of the Delphi study should be stated more clearly and have therefore elaborated in the “Scope” section. We would wish to emphasize that the Delphi process is not intended to generate inclusion or exclusion criteria but is intended to reach consensus on the most important outcome measures which can subsequently be used in future research or clinical follow-up.

It would be helpful if the authors could elaborate on the plans for the RCT as this would help the reader to conceptualize what the COS might look like. Right now it is unclear. On a related note, if the COS is going to include imaging surveillance plan for patients randomized to non-operative management, it seems like radiologists should be included in stakeholder group and I did not see them mentioned.

We agree with the reviewer that perhaps the design of the final COS may be unclear and have elaborated on this matter in the “Delphi study” section under the “Delphi round 3 analysis and final COS development” heading. We cannot provide a more detailed description of the RCT at this time as the methodology and design is to be finalized after the final COS is determined. We have included caregivers involved in the primary care and follow-up of CPAM patients in the stakeholder groups and shall include relevant specialists (ie. Radiology/Pathology) when designing the RCT.

I am very excited to see that development of an RCT to answer questions about the optimal approach to management of children with asymptomatic CPAM is underway!

Reviewer: 3

I would like to thank the editor for giving me the privilege to review the manuscript entitled:

“Development of a Core Outcome Set for Congenital Pulmonary Airway Malformations: study protocol of an international Delphi survey”.

Congratulations for a well written protocol and great effort on a very important and a debatable segment in thoracic surgery spectrum - Management of congenital pulmonary airway malformations. The working group have chosen the Delphi technique where a group of experts exchange views, and each independently gives estimates and assumptions to a facilitator who reviews the data and issues a summary report.

The working group has divided the expert teams into two groups to review the diversity in the management within surgeons and non-surgeons. The results of this study would be included in a clinical trial.

It is written using good English, I believe it is satisfactory and suitable for publication pending minor revisions.

Kindly note the following corrections I would request you to make to improve the readability of your manuscript.

1. The abstract: I recommend to a conduct systematic review of all relevant literature according to the Preferred Reporting Items for Systematic Reviews and Meta-Analyses (PRISMA) Statement. Retrospective studies and non-randomized trials are considered evidence that should be mentioned.

2. Please report your literature search strategy. Kindly mention Study selection criteria. Please report that you will only include randomized controlled trials and explain why would you exclude non RCT's and cohort studies. Thank you.

We agree with the reviewer that a systematic review of all relevant literature is an ideal method to gather relevant outcome parameters for the Delphi process. Unfortunately, no randomized trials or prospective studies have yet been published regarding the outcome of CPAM patients. We have therefore obtained outcome parameters from extensive narrative reviews which encompass all relevant literature. In addition, participants of the Delphi process have the option of suggesting additional parameters in the first round of the Delphi process which provides these experts the opportunity to provide other relevant outcome parameters.

3. All relevant outcome parameters identified from the above-mentioned reviews will be used in the first round of “the” Delphi process. Please identify and mention the role of the facilitator. In case of disagreement, please mention how would this be resolved. Thank you.

We agree with the reviewer that this role is indeed unclear and have therefore elaborated on this matter in the “Literature review” section.

4. Please mention that the outcomes will be formatted into questions and taken forward to the Delphi consensus exercise. To ensure appropriate phrasing and understanding for all stakeholders. Thank you.

We agree with the reviewer and have added this to the “Literature review” section.

5. Would the participants who registered for the survey be given a unique number to allow the Study Management Group to track attrition?

Indeed all participants will be given a unique identification number and attrition bias will be assessed as described in the “Attrition bias” section.

6. Please report the duration of each Delphi round. What is the policy of management of non-respondents? When and how would they be notified electronically?

The duration of each Delphi round is mentioned in the section “Delphi study” under the header “Timeline”:

“Participants will be asked to complete each round of the Delphi process within 4 weeks. A weekly reminder email will be sent to those who then have not yet completed the survey. Those failing to complete the questionnaire within the allocated 4 weeks will be excluded from next rounds. The

deadline shall be extended if the projected minimum sample size has not been reached and those failing to complete the questionnaire shall be approached individually.”

7. Please report data extraction method and analysis.

The data will be collected using an automated online tool “Welphi” (“Ethics and dissemination” section under the “Data collection and confidentiality” heading). Analysis of each round is separately described in the “Delphi study” section.

8. Please report the current trial status. Thank you.

The trial design will be finalized after the final COS has been determined. We have added this information in the “Scope” section.

9. Please revise the referencing style for references 17 and 18.

Thank you for your critical review; we have revised the references in question.

10. Please check 2 typos. (first round of the -- (1) Surgeons (e.i. pediatric --)

Thank you once - we have corrected spelling.

VERSION 2 – REVIEW

REVIEWER	Berman, Loren Nemours Children's Hospital
REVIEW RETURNED	22-Feb-2021

GENERAL COMMENTS	I stand by my recommendation to complete the PRISMA literature review and include its results in this publication. The description of the Delphi protocol is currently very limited and vague, and the PRISMA review will be performed in order to provide content for the Delphi panel. IF the results of this review were available for presentation along with the Delphi protocol, it would provide for a more substantive protocol. I raised this concern in my initial review ("my overall comment is that i would suggest completing the systematic literature review and including the content for Delphi method process in the protocol publication") and it was not addressed by the authors.
---

REVIEWER	Beshir, Hatem Mansoura University Faculty of Medicine, Cardiothoracic Surgery
REVIEW RETURNED	22-Feb-2021

GENERAL COMMENTS	I would like to thank the editor for giving me the privilege to review the revised version of the manuscript and the authors for their careful reading of comments on the manuscript and their constructive corrections. The authors have taken the reviewer comments on board to improve and clarify the manuscript. Thank you for your efforts on a study protocol that sets out an important and interesting issue, and is a valuable addition to congenital pulmonary malformations.
--

VERSION 2 – AUTHOR RESPONSE

Reviewer: 2

I stand by my recommendation to complete the PRISMA literature review and include its results in this publication. The description of the Delphi protocol is currently very limited and vague, and the PRISMA review will be performed in order to provide content for the Delphi panel. IF the results of this review were available for presentation along with the Delphi protocol, it would provide for a more substantive protocol. I raised this concern in my initial review ("my overall comment is that i would suggest completing the systematic literature review and including the content for Delphi method process in the protocol publication") and it was not addressed by the authors.

Indeed we did not specifically reply to this remark but have addressed it when replying to a similar comment of another reviewer: We agree with the reviewer that a systematic review of all relevant literature is an ideal method to gather relevant outcome parameters for the Delphi process. Unfortunately, no randomized trials or prospective studies have yet been published regarding the outcome of CPAM patients. We have therefore obtained outcome parameters from extensive narrative reviews which encompass all relevant literature. In addition, participants of the Delphi process have the option of suggesting additional parameters in the first round of the Delphi process which provides these experts the opportunity to provide other relevant outcome parameters.

We have added this as a limitation of the protocol in the “ Strength and limitations” section.

Reviewer: 3

I would like to thank the editor for giving me the privilege to review the revised version of the manuscript and the authors for their careful reading of comments on the manuscript and their constructive corrections. The authors have taken the reviewer comments on board to improve and clarify the manuscript. Thank you for your efforts on a study protocol that sets out an important and interesting issue, and is a valuable addition to congenital pulmonary malformations.

We thank the reviewer for the repeated effort in reviewing the manuscript and appreciate the kind words of encouragement.